# Fluorescence Spectroscopic Investigation of Competitive Interactions between Quercetin and Aflatoxin B_1_ for Binding to Human Serum Albumin

**DOI:** 10.3390/toxins11040214

**Published:** 2019-04-09

**Authors:** Hongxia Tan, Lu Chen, Liang Ma, Shuang Liu, Hongyuan Zhou, Yuhao Zhang, Ting Guo, Wei Liu, Hongjie Dai, Yong Yu

**Affiliations:** 1College of Food Science, Southwest University, Chongqing 400715, China; tanhongxia@mail.swu.edu.cn (H.T.); chenluch@email.swu.edu.cn (L.C.); liushuang9995@stu.ouc.edu.cn (S.L.); zhouhy@swu.edu.cn (H.Z.); zhy1203@swu.edu.cn (Y.Z.); guoting06@swu.edu.cn (T.G.); lwissue@email.swu.cn (W.L.); daihongjie@swu.edu.cn (H.D.); yuyong@swu.edu.cn (Y.Y.); 2Biological Science Research Center, Southwest University, Chongqing 400715, China

**Keywords:** aflatoxin B_1_, quercetin, human serum albumin, competitive interaction, fluorescence spectroscopy

## Abstract

Aflatoxin B_1_ (AFB_1_) is a highly toxic mycotoxin found worldwide in cereals, food, and animal feeds. AFB_1_ binds to human serum albumin (HSA) with high affinity. In previous experiments, it has been revealed that reducing the binding rate of AFB_1_ with HSA could speed up the elimination rate of AFB_1_. Therefore, we examined the ability of quercetin to compete with AFB_1_ for binding HSA by fluorescence spectroscopy, synchronous spectroscopy, ultrafiltration studies, etc. It was shown that AFB_1_ and quercetin bind to HSA in the same Sudlow site I (subdomain IIA), and the binding constant (K_a_) of the quercetin-HSA complex is significantly stronger than the complex of AFB_1_-HSA. Our data in this experiment showed that quercetin is able to remove the AFB_1_ from HSA and reduce its bound fraction. This exploratory work may be of significance for studies in the future regarding decreasing its bound fraction and then increasing its elimination rate for detoxification. This exploratory study may initiate future epidemiological research designs to obtain further in vivo evidence of the long-term (potential protective) effects of competing substances on human patients.

## 1. Introduction

Aflatoxin B_1_ (AFB_1_), a kind of carcinogenic toxin, which had been classified as a group I carcinogen [1] by the International Agency for Research on Cancer, is often found in agricultural products, feed, and food such as grain and oil through various channels. According to the statistics, an estimated five billion people worldwide are exposed to high levels AFB_1_, and human health is in serious dangers [2,3,4,5]. How to reduce or degrade the toxicity of AFB_1_ in agricultural production is one of the key research directions in the food safety field. Many techniques for reducing toxicity or degrading the structure of AFB_1_ have been researched and reported. For instance, the toxicity of AFB_1_ can be reduced by vitamin E through lowering the activities of plasma aspartate aminotransferase, alanine aminotransferase, and alkyne phosphatase [6]. A macromolecular complex was formed by β-D-glucan with AFB_1_ in vivo to attenuate its toxicity [7]. The toxicity of AFB_1_ could be decreased by papaya extract through decreasing the oxidative stress reaction [8]. Most detoxification in the reports was taken as the effects after the toxin reacted with the target organ [9,10,11,12]. However, there is still a very high risk of damaging the target organ by AFB_1_. Therefore, it is of great significance to study the detoxification to AFB_1_ during the transport process of AFB_1_ in vivo.

Most xenobiotics will bind and be transported by serum albumin before they react with the target organ in the blood circulation in vivo [13]. AFB_1_ follows a similar behavior as serum albumin in vivo, and many types of DNA damage can be induced by 8,9-epoxy AFB_1_, which is produced in the metabolic process of AFB_1_ through the catalytic action of the cytochrome P450 enzyme in the target organ (Figure 1) [14]. Recently, it has been reported that the time of the toxin molecules binding to serum albumin could affect the half-life of the toxin in the organism [15,16,17]. Koszegi et al. [18] reported that some flavonoids such as quercetin could compete with and displace ochratoxin A (OTA) from human serum albumin (HSA). After the competitive interaction between flavonoids and OTA to bind with HSA, it was found that the bound fraction of OTA was decreased, and then, its half-life was decreased [18]. Hence, it is speculated that the toxicity of AFB_1_ to target organs will be reduced and its elimination rate will be accelerated during the bio-transportation process due to the broken interaction, or its displacement, between AFB_1_ and HSA. It has been reported that quercetin had high affinity to HSA in the pH 7.4 condition in vitro, and the primary binding site on albumin was the same as that of the AFB_1_-HSA complex (subdomain IIA) [19,20,21,22,23]. It has been reported that quercetin is a highly-active flavonoid that regulates the uptake of toxins to reduce body damage [24]. Therefore, quercetin was thought to be a very effective competitor in the binding of AFB_1_ to HSA. 

In this proposal, the effect of quercetin on the AFB_1_-HSA complex was studied by fluorescence spectroscopy, synchronous spectroscopy, ultrafiltration studies, etc. The aim is to explore the ability of quercetin to bind competitively to HSA with AFB_1_. Our results showed that quercetin is able to remove the AFB_1_ from HSA, and this is the first time this molecular displacement has been described as a new type of interaction of quercetin affecting the biological behavior of AFB_1_.

## 2. Results 

### 2.1. Investigation of the Interaction between HSA with AFB_1_ and Quercetin

To analyze the ability of quercetin to compete with AFB_1_ for binding to HSA, we first studied the interaction mechanism of the AFB_1_-HSA system and the quercetin-HSA system, respectively. In this study, fluorescence emission spectra of albumin were recorded in the presence of increasing AFB_1_ and quercetin concentration (Figure 2). In order to exclude the inner-filter effect, emission intensities were corrected by Equation (1). In a concentration-dependent fashion, AFB_1_ and quercetin induced the decrease of fluorescence at 340 nm (emission maximum of HSA), resulting from the quenching effects of AFB_1_ and quercetin. Based on the fluorescence quenching (Figure 2), the K_sv_, K_q_, and K_a_ of the AFB_1_-HSA complex and quercetin-HSA complex were calculated (Table 1) by Equations (2) and (4). In practice, k_q_ cannot be larger than 2 × 10^10^ L mol^−1^ S^−1^ for a collisional quenching process [25,26]. We found that the static quenching mechanism occurred for the AFB_1_-HSA system (K_q_ = 1.623 × 10^12^) and the quercetin-HSA system (K_q_ = 1.83 × 10^13^), which corresponded to the results of previous studies such as [27,28] and [29]. Fluorescence quenching studies denoted the results of a static quenching mechanism for the AFB_1_-HSA and quercetin-HSA system. The K_a_ of the quercetin-HSA complex was one order of magnitude larger than the AFB_1_-HSA complex, indicating that the fluorescence quenching ability of quercetin to HSA is significantly stronger than AFB_1_ in the same condition.

To determine the binding site, site marker competitive experiments were carried out. Typically, ketoprofen and ibuprofen, as the most commonly-applied site markers of Sudlow’s site I (located on subdomain IIA) and Sudlow’s site II (located on subdomain IIIA) respectively, were applied in this experiment. Complex formation of the ligand (AFB_1_ or quercetin) with HSA resulted in the increased fluorescence signal of the ligand, and there was no fluorescence spectral interference of site markers (ketoprofen and ibuprofen as competitive probes) at the maximum emission wavelength of the ligand (AFB_1_ or quercetin) [25,30]. Therefore, displacement of the ligand from HSA by site markers could lead to the decreased fluorescence signal at 440 nm (AFB_1_) or 535 nm (quercetin) [25,30]. As we can see from Figure 3, In the AFB_1_-HSA and quercetin-HSA system, ibuprofen, a specific ligand for Sudlow’s site II, had little effect on the fluorescence intensity of HSA, indicating that ibuprofen could not substitute the AFB_1_ and quercetin. With the addition of ketoprofen, as a specific ligand for Sudlow’s site I, the fluorescence intensity of HSA in both complexes was greatly reduced, indicating that ketoprofen replaced AFB_1_ and that quercetin bound to HSA. Based on the above studies, we can conclude that AFB_1_ and quercetin bind to HSA in Sudlow’s site I (subdomain IIA), which is in good agreement with the modeling studies previously reported [25,26]. 

### 2.2. Investigation of the Competitive Interaction between Quercetin and AFB_1_ for HSA

In order to explore the competitive interaction between quercetin and AFB_1_ to bind with HSA, site marker experiments were carried out with fluorescence probes (ketoprofen and ibuprofen). Ibuprofen had no effect on the fluorescence spectra of AFB_1_ (red columns in Figure 4A) or quercetin (red columns in Figure 4B) in the AFB_1_-HSA-quercetin system, while the fluorescence signals of AFB_1_ (black columns in Figure 4A) or quercetin (black columns in Figure 4B) were decreased by the ketoprofen probe. These results indicated that AFB_1_ and quercetin could bind to HSA in the same Sudlow’s site I.

From the above experiments, both AFB_1_ and quercetin could specifically bind to a known site on HSA in Sudlow’s site I, and the K_a_ of the quercetin-HSA complex was much greater than that of the AFB_1_-HSA complex; thus, we hypothesized that the affinity of quercetin with HSA was much greater than AFB_1_ with HSA. 

For the sake of verification of this hypothesis, competitive interaction studies between quercetin and AFB_1_ for HSA were carried out. As shown in Figure 5, there was a significant difference between the system (HSA + AFB_1_) and the system (HSA+ quercetin) in the fluorescence intensity of HSA. It was confirmed that the fluorescence quenching ability of quercetin to HSA was significantly stronger than AFB_1_, and quercetin had stronger binding ability to bind with HSA compared to AFB_1_. Consequently, we further speculated that quercetin had the potential to compete for AFB_1_ from HSA, thereby hindering the binding of AFB_1_ to HSA.

### 2.3. Ultrafiltration Studies

Although quercetin and AFB_1_ bound to HSA at the same binding site and the binding ability of the quercetin-HSA complex was stronger than the complex of AFB_1_-HSA, it could not directly be proven that AFB_1_ could be substituted by quercetin in the presence of AFB_1_ and quercetin. Consequently, the investigation of competitive substitution by ultrafiltration studies was carried out, which determined whether quercetin was able to block the binding of AFB_1_ and HSA, decreasing the half-life of AFB_1_.

As shown in Figure 6, the addition of quercetin significantly increased the AFB_1_ concentration in the free state and decreased the AFB_1_ concentration in the bound state, which signified that the addition of quercetin could replace AFB_1_ in the AFB_1_-HSA complex, and the AFB_1_ was changed from the bound state to the free state.

### 2.4. Synchronous Fluorescence Studies

The information about the molecular environment near the chromophore molecules can be obtained by synchronous spectroscopy. The shift at maximum emission, corresponding to the changes of polarity around chromophores, is a useful way to evaluate the microenvironment of amino acid residues.

Figure 7, Figure 8 and Figure 9 present the peaks for the complex when Δλ = 15 nm and Δλ = 60 nm. For the AFB_1_-HSA system, the synchronous fluorescence presented a red shift of 1 nm at Δλ = 15nm and 2 nm at Δλ = 60 nm, which denoted that AFB_1_ was able to increase polarity around Tyr, as well as around Trp. For the quercetin-HSA system, the synchronous fluorescence showed a small blue shift (0.5 nm) at Δλ = 15 nm, which signified that hydrophobicity around the Tyr residue increased. However, the small red shift of 0.5 nm in the quercetin-HSA system at Δλ = 60 nm demonstrated that the hydrophobicity around the Trp residue decreased. For the AFB_1_-quercetin-HSA system, where AFB_1_ and quercetin were increasingly added and HSA was at constant concentration, a small red shift of 0.5 nm at Δλ = 60 nm was observed, but no shift when Δλ = 15 nm. These results suggested that in ternary system, ligands bind near the Trp residue. It can be inferred from the above experimental phenomena that the presence of quercetin affected the hydrophobicity of the protein microenvironment and changed the interaction of AFB_1_ with HSA. 

### 2.5. The Effect of the Ratio of Quercetin to HSA on the Competitive System

For the sake of estimating the effects of different ratios of AFB_1_: quercetin on the competitive interaction between quercetin and AFB_1_ to bind with HSA, the fluorescence intensity of HSA in different systems was examined (Table 2). Here, and in the tables below, the first column represents different system conditions. The other columns represent the value of the fluorescence intensity of HSA. As shown in the second and third lines in Table 3, the fluorescence intensity of HSA in the system (HSA + AFB_1_ + quercetin) had a significant difference from the AFB_1_-HSA system (*p* < 0.05), which indicated that the addition of quercetin significantly affected the fluorescence intensity of HSA in the AFB_1_-HSA system. In the fourth and fifth lines in Table 3, the fluorescence intensity of HSA in the system (HSA + quercetin + AFB_1_) had no significant difference from the HSA + quercetin system (*p* < 0.05), which indicated that the addition of AFB_1_ had no influence in the HSA + quercetin system. These results indicated that quercetin is able to remove AFB_1_ from HSA and that AFB_1_ is not able to remove quercetin from HSA. In the system (HSA + (AFB_1_ + quercetin)), the fluorescence intensity had a significant difference compared with the system (HSA + AFB_1_), and the fluorescence intensity had no significant difference compared with the system (HSA + quercetin). It was revealed that HSA was prone to binding to quercetin and that quercetin could prevent the binding between AFB_1_ and HSA when the molar concentration ratio of AFB_1_:quercetin was 1:2.

As shown in Table 3, the same results were presented in 1:1 system as 1:2 system, indicating that quercetin can displace AFB_1_ from HSA and that HSA was more prone to binding to quercetin, which prevented the binding of AFB_1_ to HSA.

As shown in Table 4, the fluorescence intensity of HSA in the system (HSA + AFB_1_ + quercetin) was significantly different from the system (HSA + AFB_1_) (*p* < 0.05). The fluorescence intensity of HSA in the system (HSA + quercetin + AFB_1_) was not significantly different from the system (HSA + quercetin). In the system (HSA + (AFB_1_ + quercetin)), the fluorescence intensity had a significant difference compared with the system (HSA + AFB_1_), and the fluorescence intensity had no significant difference compared with the system (HSA + quercetin). These results revealed that quercetin is still able to competitively bind to HSA with AFB_1_ when the ratio of AFB_1_:quercetin is 2:1.

## 3. Discussion

In previous experiments, several mycotoxins’ ligands (including AFB_1_, OTA, citrinin (CIT), deoxynivalenol (DON), patulin, zearalenone (ZEN), etc. [27,31,32]) formed stable complexes with HSA. HSA forms non-covalent complexes with these ligands, which could significantly affect the distribution and elimination of these ligand molecules [27,31,32,33,34], Especially, OTA binds primarily to albumin with high affinity (K_a_~10^7^ L/mol) [18], which gives rise to its longer plasma elimination half-life (from a few days to one month) [22,35,36]. Sigrid et al. [36] studied the toxicokinetic profile of OTA (50 ng/g) after the oral or intravenous administration to fish, quail, mouse, rat, and monkey. It was found that the elimination of OTA can be completed through renal and hepatic clearance, both of which can be influenced by the toxin’s binding to plasma macromolecules. Irene et al. [37] assessed the toxicokinetic profile of OTA by one human volunteer. This half-life of OTA was approximately eight-times longer than that determined previously in rats, which accounted for the specific OTA binding plasma protein species and respective protein concentrations. Furthermore, Kumagai et al. [16] studied the changes of OTA in both albumin-deficient and normal rats. It was revealed that the concentration of OTA in plasma was reduced to <0.5 μg/mL within 10 min of the injection in albumin-deficient rats, while remaining >50 μg/mL for 90 mins in normal rats [16]. The concentrations of OTA in bile and urine, as well as the excretion rate of OTA in these fluids were 20–70-fold higher in albumin-deficient than in normal rats [16]. In summary, it was demonstrated that the primary effect of albumin binding on OTA is to retard its elimination by restricting the entry of OTA into the hepatic and renal cells [16]. 

It is well known that natural flavonoids can also bind to HSA at the same binding site as OTA does (site I, subdomain IIA). Miklós Poór et al. [18] studied the competitive interaction between flavonoids and OTA, which found that some flavonoid aglycones are able to remove the toxin from HSA and decrease the bound fraction of OTA. Baudrimonta et al. [17] had found that piroxicam also can bind strongly to plasma proteins and could stop OTA binding with HSA and transporting to target organs. Miklós Poór et al. [24] studied the competitive interactions between OTA and 13 drug molecules for binding to HSA, which demonstrated that some drugs show high competitive capacity. In addition, several extracorporeal dialysis procedures using albumin-containing dialysates have proven to be an effective tool for removing endogenous toxins or overdosed drugs from patients [38,39,40].

As we all know, AFB_1_ can bind to HSA with high affinity (K~10^4^ L/mol). The high affinity between AFB_1_ and HSA could affect its distribution and elimination [27]. It has not been studied at present whether AFB_1_ is removed from HSA and then reduces the bound fraction by competitive interaction. Besides, flavonoids could bind to HSA with high affinity (K~10^5^ L/mol) at the same binding site as AFB_1_ does (site I, subdomain IIA) [41,42]. Especially, quercetin is one of the most common flavonoids in nature and can bind with human albumin with high affinity [43]. Furthermore, quercetin was the most effective competitor of OTA in the competition experiment between OTA and polyphenols [18]. Therefore, quercetin was adopted to study the competitive interaction between AFB_1_ and quercetin when binding with HSA. 

However, no data were found for any trial using quercetin in competitive AFB_1_-HSA models. In this experiment, fluorescence quenching studies denoted the results of a static quenching mechanism for the AFB_1_-HSA and quercetin-HSA (Figure 2) complex. The K_a_ of quercetin-HSA was one order of magnitude larger than AFB_1_-HSA (Table 1), indicating that the fluorescence quenching ability of quercetin to HSA is significantly stronger than AFB_1_ in the same condition. By the competitive probe experiment, we can conclude that Sudlow’s site I is a high affinity binding site of the AFB_1_-HSA and quercetin-HSA complex, which is in good agreement with modeling studies previously reported [25,26]. Quercetin and AFB_1_ bind to HSA at the same binding site (Figure 3); thus, it is possible that quercetin and AFB_1_ could competitively bind with HSA. For the sake of verification of the above hypothesis, research of the competitive interaction between quercetin and AFB_1_ to bind with HSA was carried out (Figure 4 and Figure 5). It was revealed that in the competition system, the binding sites of quercetin and AFB_1_ on HSA were still on Sudlow’s site I. The binding ability of quercetin with HSA was significantly stronger than that of AFB_1_-HSA, which indicated that quercetin had the ability to replace HSA from AFB_1_. By ultrafiltration studies (Figure 6), it was further confirmed that quercetin was able to remove AFB_1_ from HSA and then decreased the bond fraction of AFB_1_. 

According to other published papers, it was found that quercetin could affect the AFB_1_-induced negative changes. Choi et al. [44] found that quercetin does not directly protect against AFB_1_-mediated liver damage in vivo, but plays a partial role in promoting antioxidative defense systems and inhibiting lipid peroxidation. Additionally, El-Nekeety et al. [45] found that quercetin has potential antioxidant activity and could regulate the alteration of genes expression induced by AFB_1_. However, in this experiment, based on the competitive interaction between quercetin and AFB_1_ binding to HSA, respectively, we can conclude that quercetin is able to remove AFB_1_ from HSA and decrease the bound fraction of AFB_1_. The effect of quercetin on AFB_1_-induced negative changes was studied in this experiment from different aspects compared with the above [9,10,11,44,45] studies.

## 4. Conclusion

Competitive interaction between quercetin and HSA was investigated by fluorescence spectroscopy, synchronous spectroscopy, ultrafiltration studies, etc. This is the first time evidence that the addition of quercetin can remove AFB_1_ from HSA and decrease the bound fraction of AFB_1_ has been illustrated. Furthermore, experiments in vivo are necessary in the future to explore the toxicological outcome of the competitive interaction between AFB_1_ and quercetin with HSA. 

## 5. Materials and Methods

### 5.1. Reagents

Aflatoxin B_1_ (AFB_1_, from Sigma), human serum albumin (HSA, from Sigma), quercetin (from Aladdin), ketoprofen (from Shanghai chemical Technology Co., ltd), and ibuprofen (Beijing Bailingwei Technology Co., ltd) were used as received. Tris solution was obtained from Amersco. Other chemicals were of analytical grade.

### 5.2. Fluorescence Spectroscopic Measurements

Fluorescence measurements were carried out employing an F-2500 fluorescence spectrophotometer (Shimadzu, Japan) at room temperature (25 ℃), with a 1-cm path length quartz cell, using an excitation wavelength of 280 nm. In the AFB_1_-HSA system, the concentration of HSA was fixed at 5 μM, whereas the AFB_1_ concentration was varied from 0 to 9 μM. In the quercetin-HSA system, the concentration of HSA was fixed at 5 μM, whereas the quercetin concentration was varied from 0 to 4 μM. In competitive interaction between AFB_1_ and quercetin with HSA, the concentration of HSA was fixed at 5 μM, whereas the concentration of AFB_1_ and quercetin was varied as above. In competitive interaction studies, the construction methods of different systems were as follows: The system (HSA + AFB_1_) was constructed by the addition of AFB_1_ to the HSA solution. The system (HSA + quercetin) was constructed by the addition of quercetin to the HSA solution. For the system (HSA + AFB_1_ + quercetin), AFB_1_ was added to the HSA solution firstly, and quercetin was then added to the solution. For the system (HSA + quercetin + AFB_1_), quercetin was added to the HSA solution firstly, followed by the addition of AFB_1_. For the HSA + (AFB_1_ + quercetin) system, AFB_1_ and quercetin were added to the Tris solution firstly, followed by the addition of HSA.

For the sake of eliminating the inner-filter effects, the fluorescence intensities were corrected applying the following equation [26,46]:(1)$$F_{\mathrm{cor}}=F_{\mathrm{obs}}\times e^{\frac{A_{\mathrm{ex}}+A_{\mathrm{em}}}{2}}$$
where F_cor_ and F_obs_ represent the corrected and observed fluorescence intensities, respectively; whereas A_ex_ and A_em_ denote the absorbance values at excitation and emission wavelength, respectively. The corrected fluorescence data were used for further analysis, related to HSA fluorescence quenching.

The fluorescence quenching data are usually evaluated via the Stern–Volmer equation [16,17]:(2)$$\frac{F_{0}}{F}=1+K_{q}\tau_{0}\left[Q\right]=1+K_{\mathrm{sv}}\left[Q\right]$$
where F_0_ and F are the fluorescence emission intensity without and with the addition of a known concentration [Q], respectively. K_sv_ is the Stern–Volmer constant; K_q_ is the quenching rate of the biomolecule; *τ*_0_ is the average fluorescence lifetime of HSA without quencher.

For static quenching, the modified Stern–Volmer equation analyzes the data [47,48,49]:(3)$$\frac{F_{0}}{\Delta F}=\frac{F_{0}}{F_{0}-F}=\frac{1}{f_{a}K_{a}\left[Q\right]}+\frac{1}{f_{a}}$$
where ΔF = F_0_ − F is the difference in fluorescence intensity before and after the addition of the quencher at concentration [*Q*], *f_a_* represents the fraction of accessible fluorescence, and K_a_ denotes the effective quenching constant for the accessible fluorophores [50,51].

The binding constant (K_a_) and the number of binding sites (n) can be evaluated using the following equation [52,53]:(4)$$\log\left(\frac{F_{0}-F}{F}\right)={\mathrm{logK}}_{a}+\mathrm{nlog}\left[Q\right]$$
where K_a_ is the binding constant of the interaction between the quencher and HSA and n is the number of binding sites. Based on a plot of log[(F_0_-F)/F] versus log[Q], the n equal to the slope and K_a_ can be obtained.

### 5.3. Synchronous Fluorescence

The synchronous fluorescence spectra were collected by simultaneous scanning of the excitation and emission monochromators on an F-2500 fluorescence spectrophotometer (Shimadzu, Japan). The experimental condition was that the wavelength interval (Δλ) of Thr and Trp were 15 nm and 60 nm, respectively [54,55], at 25 ℃. The concentration of HSA was 4 × 10^−6^ M. The concentrations of AFB_1_ and quercetin both were 0–8 × 10^−6^ M.

### 5.4. Site Marker Competitive Experiments

Site marker competitive experiments were performed with ketoprofen (as the site I marker) and ibuprofen (as the site II marker) [56,57]. First, 1.0 mL of the 4 × 10^−6^ M HSA solution was added to a 1-cm fluorescence cuvette, with the addition of 4 × 10^−4^ M quercetin and 4 × 10^−4^ M AFB_1_ solution to make the ratio of HSA to ligand concentration 1:1. Then, 4 × 10^−3^ M ketoprofen or ibuprofen were added the above solution, and the concentrations of ibuprofen and ketoprofen were 8.0, 16.0, 24.0, 32.0, 40.0, 48.0, 56.0, 64, and 72.0 × 10^−6^ M. 

### 5.5. Ultrafiltration

Quercetin, AFB_1_, and HSA solution was prepared at a concentration ratio of 1:1:1 and reacted at room temperature for 30 min. After the reaction, the solution was centrifuged at 4 °C at 10,000 rpm for 15 min and washed 3 times on the filter with Tris-HCL buffer, collecting the filtrated solution for further analysis of the next term. Methanol/water (50:50; v/v) was added to the filter, and the blended mixture was left standing for 30 min. Then, it was centrifuged at 4 °C at 10,000 rpm for 15 min. Here, the centrifugation operation should be repeated 3 times. Furthermore, the elution solution was collected for the fluorescence measurements. The fluorescence emission spectra in the range of 400–500 nm of free solution and the elution were scanned at an excitation of 365 nm.

### 5.6. Statistical Analyses

All of the data were statistically analyzed [53,58] by the one-way ANOVA test (IBM SPSS Statistics 20), with the level of significance at a minimum of *p* < 0.05 and a maximum of *p* < 0.01.

## Figures and Tables

**Figure 1 toxins-11-00214-f001:**
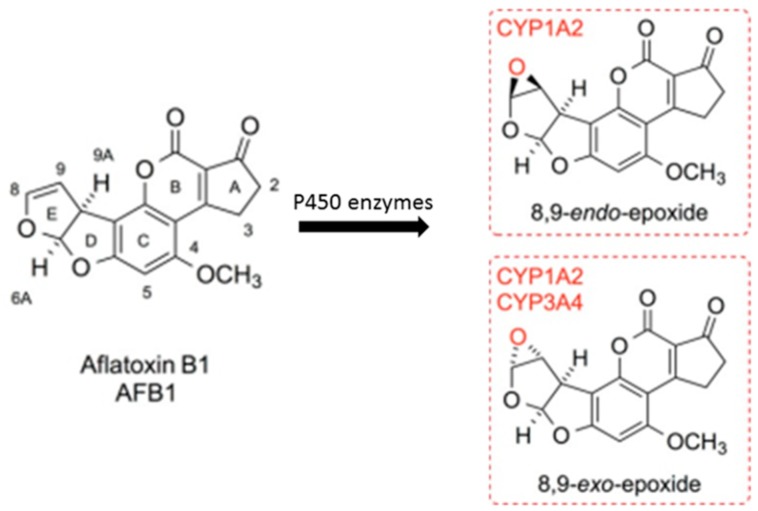
Structure of AFB_1_ and its metabolites.

**Figure 2 toxins-11-00214-f002:**
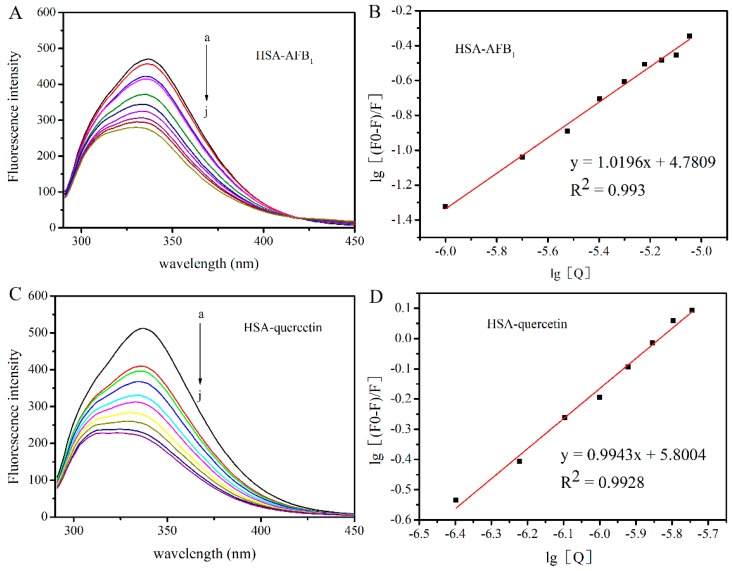
(**A**) Fluorescence emission spectra of HSA (5 μM) in the presence of increasing AFB_1_ concentration. The concentration of AFB_1_ was 0, 1, 2, 3, 4, 5, 6, 7, 8, 9 μM from a–j. (inset: the Stern–Volmer plot of the AFB_1_-HSA system). Conditions: T = 25 °C, pH = 7.4, λ_ex_ = 280 nm. (**B**) Plots of log[(F_0_ – F)/F] versus log[Q] for the interaction of HSA and AFB_1_. (**C**) Fluorescence emission spectra of HSA (5 μM) in the presence of increasing quercetin concentration. The concentration of quercetin was 0.4, 0.8. 1.2, 1.6., 2.0, 2.4, 2.8, 3.2, 3.6, 4.0 μM from a–j. (inset: the Stern–Volmer plot of the quercetin-HSA system). Conditions: T = 25 °C, pH = 7.4, λ_ex_ = 280 nm. (**D**) Plots of log[(F_0_-F)/F] versus log[Q] for the interaction of HSA and quercetin.

**Figure 3 toxins-11-00214-f003:**
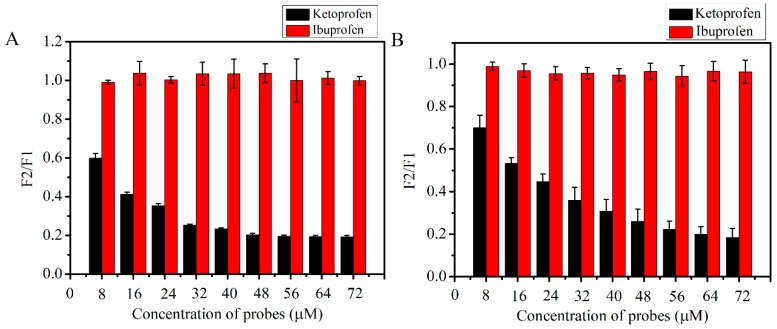
(**A**) Effects of probes on the fluorescence intensity of AFB_1_ in the system of AFB_1_-HSA. (λex = 365 nm, λem = 440 nm). (**B**) Effects of probes on the fluorescence intensity of quercetin in the system of quercetin-HSA. (λex = 455 nm, λem = 535 nm). F1: Fluorescence intensity of AFB_1_ or quercetin in the absence of probes. F2: Fluorescence intensity of AFB_1_ or quercetin in the presence of probes.

**Figure 4 toxins-11-00214-f004:**
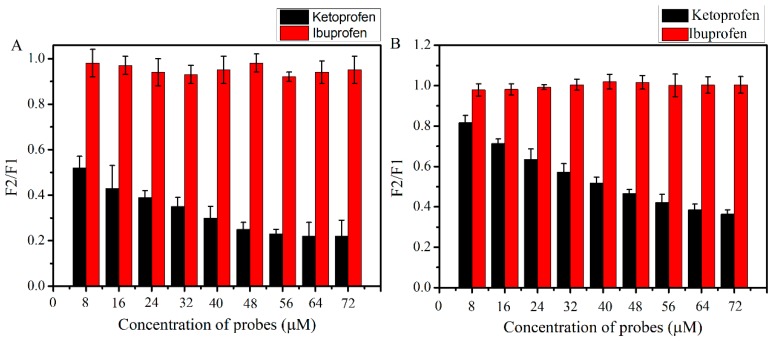
(**A**) Effects of the probes on the fluorescence intensity of AFB_1_ in the system of AFB_1_-HSA-quercetin (λex = 365 nm, λem= 440 nm). (**B**) Effects of probes on the fluorescence intensity of quercetin in the system of AFB_1_-HSA-quercetin (λex = 455 nm, λem= 535 nm). F1: Fluorescence intensity of AFB_1_ or quercetin in the absence of probes. F2: Fluorescence intensity of AFB_1_ or quercetin in the presence of probes.

**Figure 5 toxins-11-00214-f005:**
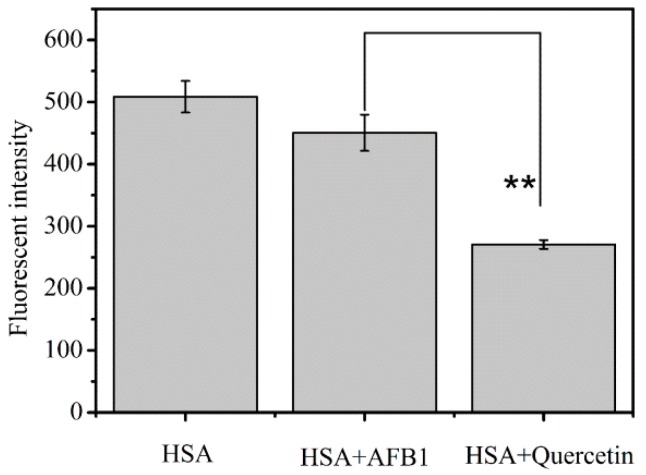
Fluorescence intensity of HSA in the absence or the presence of the AFB_1_ and quercetin (λ_ex_ = 280 nm, C_AFB1_ = C_quercetin_ = 4 × 10^−6^ M, **: *p* < 0.01 between HSA-AFB_1_ system and HSA-quercetin system).

**Figure 6 toxins-11-00214-f006:**
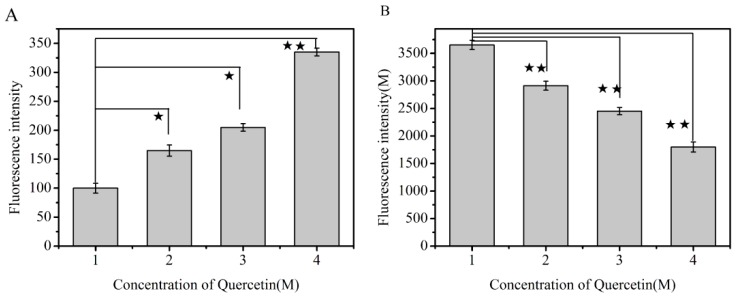
Fluorescent intensity of AFB_1_ in different systems (λ_ex_ = 365 nm, λem = 440 nm, (**A**) AFB_1_ in the free state; (**B**) AFB_1_ in binding state; *: *p* < 0.05 compared to another quercetin concentration; **: *p* < 0.01 compared to another quercetin concentration).

**Figure 7 toxins-11-00214-f007:**
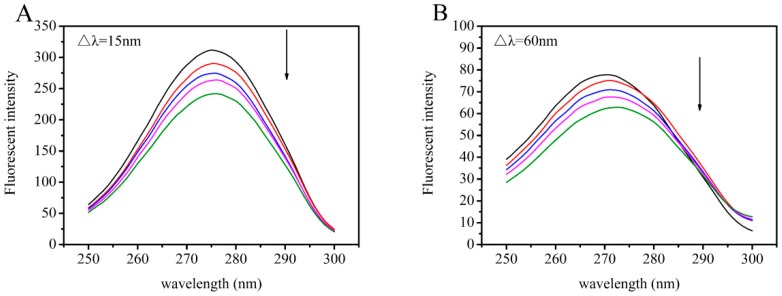
(**A**) Synchronous fluorescence spectra of the AFB_1_-HSA system, Δλ = 15 nm; (**B**) synchronous fluorescence spectra of the AFB_1_-HSA system, Δλ = 60 nm. The concentration of HSA was 4 μM, and the concentrations of AFB_1_ were 0, 2, 4, 8, 12 μM from top to bottom.

**Figure 8 toxins-11-00214-f008:**
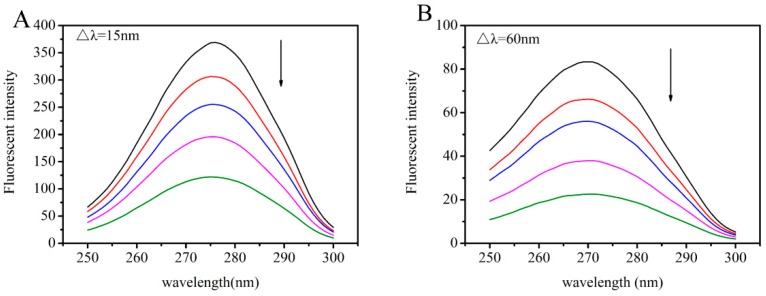
(**A**) Synchronous fluorescence spectra of the quercetin-HSA system, Δλ = 15 nm; (**B**) synchronous fluorescence spectra of the quercetin-HSA system, Δλ = 60 nm. The concentration of HSA was 4 μM, and the concentrations of quercetin were 0, 2, 4, 8, 12 μM from top to bottom.

**Figure 9 toxins-11-00214-f009:**
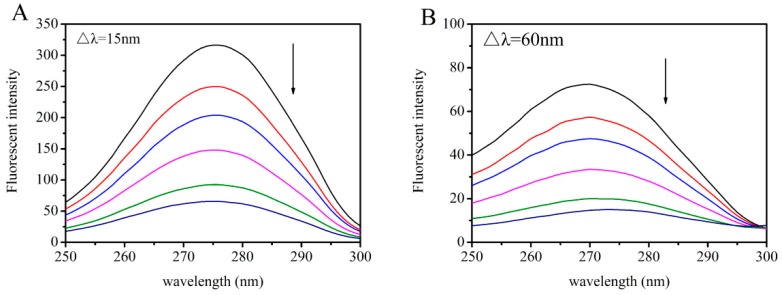
(**A**) Synchronous fluorescence spectra of the competitive interaction between quercetin and AFB_1_ for HSA, Δλ = 15 nm. (**B**) Synchronous fluorescence spectra of the competitive interaction between quercetin and AFB_1_ for HSA, Δλ = 60 nm. The concentration of HSA was 4 μM. The concentrations of quercetin and AFB_1_ were 0, 2, 4, 8, 12 μM from top to bottom.

**Table 1 toxins-11-00214-t001:** The values of K_sv_, K_q_, K_a_, and the possible number of binding sites (n) for various systems are estimated.

System	K_sv_ (LmoL^−1^)	K_q_ (LmoL^−1^S^−1^)	K_a_ (M^−1^)	n	R^2^
AFB_1_-HSA	1.62 × 10^4^	1.62 × 10^12^	6.02×10^4^	1	0.993
Quercetin-HSA	1.83 × 10^5^	1.83 × 10^13^	6.31×10^5^	1	0.992

**Table 2 toxins-11-00214-t002:** The fluorescence intensity of HSA in different systems when the molar concentration rate of AFB_1_: quercetin is 1:2 (λex = 280 nm, λem = 345 nm).

Systems	HSA:AFB_1_:Quercetin	
	2:1:2	1:1:2
HSA	494.3 ± 41.6 ^a^	498.7 ± 61.8 ^a^
HSA + AFB_1_	424.7 ± 58.5 ^b^	444.9 ± 19.4 ^b^
HSA + AFB_1_ + Quercetin	251.4 ± 17.8 ^d^	282.7 ± 37.9 ^c^
HSA + Quercetin	292.6 ± 31.0 ^b, c^	319.8 ± 31.3 ^b^
HSA + Quercetin + AFB_1_	254.4 ± 22.9 ^c^	265.4 ± 38.5 ^b^
HSA + (AFB_1_ + Quercetin)	315.6 ± 20.7 ^c^	281.8 ± 49.9 ^c^

Note: The numbers marked by the same letters (“a”, “b”, “c”, and “d”) represent the existence of statistical significance (*p* < 0.05) in the same column.

**Table 3 toxins-11-00214-t003:** The fluorescence intensity of HSA in different systems when the molar concentration rate of AFB_1_: quercetin is 1:1 (λex = 280 nm, λem = 345 nm).

Systems	HSA:AFB_1_:Quercetin	
	2:1:1	1:1:1
HSA	512.8 ± 5.7 ^a^	530.9 ± 57.8 ^a^
HSA + AFB_1_	433.6 ± 15.2 ^b^	497.1 ± 23.9 ^b^
HSA + AFB_1_ + Quercetin	376.3 ± 14.3 ^c^	375.7 ± 64.1 ^c^
HSA + Quercetin	374.1 ± 16.7 ^b^	407.3 ± 21.6 ^b^
HSA + Quercetin + AFB_1_	360.5 ± 27.5 ^b^	386.7 ± 64.2 ^b^
HSA + (AFB_1_ + Quercetin)	398.6 ± 19.9 ^c^	414.1 ± 23.4 ^c^

Note: The numbers marked by the same letters (“a”, “b”, and “c”) represent the existence of statistical significance (*p* < 0.05) in the same column.

**Table 4 toxins-11-00214-t004:** The fluorescence intensity of HSA in different systems when the molar concentration rate of AFB_1_: quercetin is 2:1 (λex = 280 nm, λem = 345 nm).

Systems	HSA:AFB_1_:Quercetin	
	2:2:1	1:2:1
HSA	467.3 ± 40.9 ^a^	501.8 ± 31.2 ^a^
HSA + AFB_1_	405.7 ± 42.9 ^b^	473.8 ± 2.7 ^b^
HSA + AFB_1_ + Quercetin	315.9 ± 21.6 ^c^	37.3 ± 19.4 ^c^
HSA + Quercetin	328.0 ± 21.6 ^b^	365.5 ± 7.6 ^b^
HSA + Quercetin + AFB_1_	362.9 ± 30.4 ^b^	72.0 ± 43.4 ^b^
HSA + (AFB_1_ + Quercetin)	362.9 ± 30.4 ^b^	72.0 ± 43.4 ^b, c^

Note: The numbers marked by the same letters (“a”, “b”, and “c”) represent the existence of statistical significance (*p* < 0.05) in the same column.

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
