# Peer review of "Fluorescence Spectroscopic Investigation of Competitive Interactions between Quercetin and Aflatoxin B1 for Binding to Human Serum Albumin"

_toxins, 2019, doi:10.3390/toxins11040214_

Round 1

Reviewer 1 Report

General comments:

1) Authors present a paper with new data on the competition between AFB1 and quercetin for binding site at human serum albumin. Though these data are undoubtedly new and elucidate the mechanism of protective quercetin action, I would suggest authors to mention (either in the Introduction, or Discussion sections) the previous studies of other authors about the quercetin effect on AFB1-induced negative changes in the organism (see below). Additionally, in my opinion, these earlier studies do not allow the authors of this study to conclude that their paper provides “new ideas for detoxification” (Conclusions section) – see the comment below.

2) Good level of English, so the text is quite understandable, but some language editing would improve the situation.

3) Please, check the text for missed spaces, especially between the references ([1], [2], etc.) and the preceding word, as well as between the words (for example, line 28: Amacromolecular…. The absence of spaces makes the reading to be more difficult.

Other comments:

Fig. 1: What do vertical arrows indicate in Fig. 1A and C? Please, add explanation to the legend.

Line 92: how did you prepare AFB1-quercetin-HSA complex? There is no information in “materials and methods” section.

Fig. 5: What do two asterisks above the HAS + Quertecin bar mean? The legend contains double repetition of “the same concentration” phrase.

Fig. 6: What do the asterisks on figures mean?

Line 154: “In the AFB1:quercetin=1:2 system, the addition of quercetin…“ do you mean AFB1:HSA system?

Table 3: Please, indicate in the table, what are the data shown in it (fluorescence intensity, wavelength) and also what do the letter a, b, and c indicate. Both figures and tables should be self-explanatory. Also, what does the first column (HAS, HAS+ABFB1, etc.) mean? It is difficult to understand; probably, the design of the table should be somehow rearranged.

The same comments are also for the next two tables.

Line 177: please, replace HAS to HAS.

Lines 193-194: “Based on previous research and the results of this experiment, 193 we can conclude that some flavonoids such as quercetin could regulates the uptake of toxins to reduce 194 body damage. In addition, this article also provides new ideas for detoxification” – Actually there are some studies showing that quercetin provides some protective effect against AFB1, so you result can not be considered as a new idea for detoxifixation. See, for example:

https://www.sciencedirect.com/science/article/pii/S0278691510004400?via%3Dihub

https://www.sciencedirect.com/science/article/pii/S2214750014000353  

Though these studies described changes at the level of activity of some enzymes, while your study deals with the details of competitive interaction at a HSA binding site, I wander why you did not mention these papers in the discussion or introduction sections. In my opinion, it should be done.

Line 198: “remove quercetin is…” probably you meant “addition of quercetin …”?

Author Response

Dear editor,

According to the comments of reviewers, a drastic revision of my manuscript was done. All of the changes were marked in the revised manuscript in red. The format, spelling, language, information of authors, and etc. in manuscript have also been revised. The details about revised resubmission were the followings.

I marked these changes in colored red.

For Reviewer #1

(1). Authors present a paper with new data on the competition between AFB1 and quercetin for binding site at human serum albumin. Though these data are undoubtedly new and elucidate the mechanism of protective quercetin action, I would suggest authors to mention (either in the Introduction, or Discussion sections) the previous studies of other authors about the que rcetin effect on AFB1-induced negative changes in the organism (see below). Additionally, in my opinion, these earlier studies do not allow the authors of this study to conclude that their paper provides “new ideas for detoxification” (Conclusions section) – see the comment below.

Thanks for your comments. We agree with your suggestion. We have supplemented clarifications on this point in discussion section and conclusion section in Line 268-308, 320-336 and 352-356.

In line 268-308, 320-336 and 352-356, we added some analysis about the Quertecin effect on AFB1-induced negative changes in the organism through the literature survey.

The half-time of toxins could be shorter with the reduction of the bound fraction of toxins to HSA [32], [33], [34], [35]. Meanwhile, the inappropriate statements have been revised in manuscript.

(2). Good level of English, so the text is quite understandable, but some language editing would improve the situation.

Thank you for your positive evaluation on the language. And for further language improvement, we have made some changes in revised manuscript. We marked the changes in red in revised manuscript.

(3). Please, check the text for missed spaces, especially between the references ([1], [2], etc.) and the preceding word, as well as between the words (for example, line 28: Amacromolecular…. The absence of spaces makes the reading to be more difficult.

Thank you for your reviews. We agree with your suggestion. We revised the wrong spelling, missed spaces, and kinds of nonstandard format.

(4). Fig. 2: What do vertical arrows indicate in Fig. 1A and C? Please, add explanation to the legend.

Thanks for your comments. We added the notes (“a” to “j”) for the vertical arrows in Fig. 2A and C. The vertical arrows indicate that the concentration of AFB1 was 0, 1, 2, 3, 4, 5, 6, 7, 8, 9 μM from “a” to “j”.

(5). Line 92: how did you prepare AFB1-quercetin-HSA complex? There is no information in “materials and methods” section.

Thank you for your comments. We found that it was not so precise for the “AFB1-quercetin-HSA complex”. The name of “complex” should be replaced by “complex system”. We have supplemented some explanation in materials and methods section in Line 366-371.

Line 366-371: In AFB1-HSA system, the concentration of HSA was fixed at 5 μM, whereas AFB1 concentration was varied from 0⁓9 μM. In quercetin-HSA system, the concentration of HSA was fixed at 5 μM, whereas quercetin concentration was varied from 0⁓4 μM. In competitive interaction between AFB1 and quercetin with HSA, the concentration of HSA was fixed at 5 μM, whereas the concentration of AFB1 and quercetin were varied as above.

(6). Fig. 5: What do two asterisks above the HSA + Quertecin bar mean? The legend contains double repetition of “the same concentration” phrase.

Here, the two asterisks indicate that there was a significant difference at the same concentration of AFB1 and quercetin on the fluorescence quenching of HSA (p < 0.01). We have supplemented clarifications on this point in our revised manuscript in Line 157-158.

(7). Fig. 6: What do the asterisks on figures mean?

Thanks for your reviews. The two asterisks indicate that there was a significant difference between different quercetin concentration (p < 0.01). The one asterisks indicate that there a difference between different quercetin concentration (p < 0.05).

We have supplemented clarifications on this point in our revised manuscript in Line 172-173.

(8). Line 154: “In the AFB1:quercetin=1:2 system, the addition of quercetin…“ do you mean AFB1:HSA system?

Yes, it is. Here we studied the effects of the addition of quercetin on the fluorescence intensity of HSA in AFB1-HSA system (the final ratio of AFB1:quercetin=1:2). We have made some supplement in Line 209-223.

(9). Table 3: Please, indicate in the table, what are the data shown in it (fluorescence intensity, wavelength) and also what do the letter a, b, and c indicate. Both figures and tables should be self-explanatory. Also, what does the first column (HAS, HAS+ABFB1, etc.) mean? It is difficult to understand; probably, the design of the table should be somehow rearranged.

Thanks for your suggestions. We have made some supplementary explanation in the note of Table 3 (Line 209-211 and 236-237) and methods section in Line 371-378.

Line 236-237: Different letters (a, b, c, d) represent the existence of statistical significance (p < 0.05) in the same column.

Line 209-211: In Table 3, the 1st and 4th columns represented different system conditions. The other columns represented the value of fluorescence intensity of HSA. Line 371-378: In competitive interaction studies, the construction methods of different systems are as follows: The system (HSA+AFB1) was constructed by the addition of AFB1 to HSA solution. The system (HSA + Quercetin) was constructed by the addition of quercetin to HSA solution. The system (HSA + AFB1 +quercetin) was that AFB1 was added to the HSA solution firstly, and quercetin was then added to the solution. The system (HSA + Quercetin + AFB1) was that quercetin was added to HSA solution firstly, followed the addition of AFB1. The HSA + (AFB1 +Quercetin) system was that AFB1 and quercetin were added to the tris solution firstly, followed the addition of HSA.

(10). The same comments are also for the next two tables.

Thanks for your comments. We have made some supplementary explanation in the note of Table 4 and 5 in Line 238-239, 246-247, 248-249 and 265-266 and methods section in Line 371-378.

(11). Line 177: please, replace HAS to HSA.

Thank you for your comment, I have changed in Line 311. 

(12). Lines 193-194: “Based on previous research and the results of this experiment, 193 we can conclude that some flavonoids such as quercetin could regulates the uptake of toxins to reduce 194 body damage. In addition, this article also provides new ideas for detoxification” – Actually there are some studies showing that quercetin provides some protective effect against AFB1, so you result can not be considered as a new idea for detoxifixation. See, for example:

https://www.sciencedirect.com/science/article/pii/S0278691510004400?via%3Dihub

https://www.sciencedirect.com/science/article/pii/S2214750014000353  

Thanks for your reviews. We have made some changes and supplemented some explanation on this point in discussion section in Line 327-336.

Line 327-336: In previous studies, it was found that quercetin could affect the AFB1-induced negative changes. Choi et al [48] found that quercetin does not directly protect against AFB1-mediated liver damage in vivo, but plays a role in promoting antioxidative defense systems and inhibiting lipid peroxidation to some degrees. El-Nekeety et al [49] found that quercetin has a potential antioxidant activity and could regulate the AFB1-induced alterations of gene expression. However, in this experiment, based on the competitive interaction between quercetin and AFB binding to HSA, respectively, we can conclude that quercetin is able to remove AFB1 from HSA and decrease the bound fraction of AFB1. The effect of quercetin on AFB1-induced negative changes was studied in this experiment from different aspects compared with above [48], [49] studies.

(13) Though these studies described changes at the level of activity of some enzymes, while your study deals with the details of competitive interaction at a HSA binding site, I wander why you did not mention these papers in the discussion or introduction sections. In my opinion, it should be done.

Thank you for your reviews. In Line 48-49 and 336, we have supplemented clarifications on this point in our revised manuscript in discussion.

(14). Line 198: “remove quercetin is…” probably you meant “addition of quercetin …”?

Thanks for your reviews. We have edited this sentence in Line 351.

Line 351: It is the first time to illustrate the evidence that the addition of quercetin can remove AFB1 from HSA and decrease the bound fraction of AFB1.

Reviewer 2 Report

The manuscript describes the studies carried out to evaluate competing binding of quercetin and AFB1 with albumin, with the aim of propose quercetin as an agent to speed up elimination of AFB1 and reduce its toxicity. As mention by the authors, previous studies have been published to study the effect of flavonoids over the binding of other mycotoxins to albumin. Therefore, the novelty of the study and the experimental dosing is limited. Even so, the topic is worthy investigating and can be of interest for the readers of Toxins.

Please find my comments below:

-          English should be reviewed through the entire document. Some sentences are incomplete or difficult to understand.

-          Abstract should be modified to reflect better what the study is about, how it was performed and the results obtained.

-          Authors should give explanation why other flavonoids where not included on the scope of the study and why quercetin is expected to provide the best effect.

-          Key contribution: please eliminate “this study demonstrated that quercetin could attenuate the damage of AFB1 to body by reducing the binding rate of AFB1 and HAS” – the study demonstrate that AFB1 binds HSA stronger than AFB1. The rest is speculation.

-          Conclusions:  should be modified to reflect better the outcome of the study carried out, without speculative sentences. Please delete “it can shorten the half-life of AFB1 in vivo and reduce its long-term high concentration of target organs…” – this has not been studied and proved.

Author Response

Dear editor,

According to the comments of reviewers, a drastic revision of my manuscript was done. All of the changes were marked in the revised manuscript in red. The format, spelling, language, information of authors, and etc. in manuscript have also been revised. The details about revised resubmission were the followings.

I marked these changes in colored red.

For Reviewer #2:

The manuscript describes the studies carried out to evaluate competing binding of quercetin and AFB1 with albumin, with the aim of propose quercetin as an agent to speed up elimination of AFB1 and reduce its toxicity. As mention by the authors, previous studies have been published to study the effect of flavonoids over the binding of other mycotoxins to albumin. Therefore, the novelty of the study and the experimental dosing is limited. Even so, the topic is worthy investigating and can be of interest for the readers of Toxins.

Please find my comments below:

(1). English should be reviewed through the entire document. Some sentences are incomplete or difficult to understand.

Thanks for your suggestions. We agree with your suggestion. We have changed some sentences in revised manuscript such as Line 59-63, 72-76, 92-96, 118, 125-127, 134-135, 148-150.

(2). Abstract should be modified to reflect better what the study is about, how it was performed and the results obtained.

Thank you for your comments. We have made some changes on this point in our revised manuscript in abstract in Line 15-21 and Line 22-24.

Line 15-21: Therefore, we examined the competitive ability of quercetin that quercetin is able to remove AFB1 from HSA and then reduce its bound fraction of AFB1 by fluorescence spectroscopy, synchronous spectroscopy, ultrafitration studies, and etc.

Line 22-24: This study may have significance for studies in the future that decreasing it bound fraction and increasing its elimination rate to detoxification.

(3). Authors should give explanation why other flavonoids where not included on the scope of the study and why quercetin is expected to provide the best effect.

Thank you for your comments. We have added some explanation on this on this point in our revised manuscript in Line 299-308. The more detailed explanations are as follows.

In the preliminary experiment, the interaction of polyphenols (e.g. luteolin, myricetin, quercetin, and etc.) with HSA was studied and it was found that quercetin has the highest affinity (Ka ~105 L/mol) with HSA. It was shown that quercetin was of the strongest competition action. Therefore, in this experiment, quercetin was selected as a representative substance of polyphenols to study the ability of competitive binding with HSA.

(4). Key contribution: please eliminate “this study demonstrated that quercetin could attenuate the damage of AFB1 to body by reducing the binding rate of AFB1 and HAS” the study demonstrates that AFB1 binds HSA stronger than AFB1. The rest is speculation.

We did some modifications in our revised manuscript in Line 29-32.

Line 29-32: It is the first time to illustrate the evidence that the addition of quercetin can remove AFB1 from HSA and reduce the bound fraction of AFB1.

(5). Conclusions: should be modified to reflect better the outcome of the study carried out, without speculative sentences. Please delete “it can shorten the half-life of AFB1 in vivo and reduce its long-term high concentration of target organs…” – this has not been studied and proved.

Thanks for your reviews. We have deleted this sentence and supplemented some sentences in conclusion in Line 352-356.

Line 352-356: Furthermore, further experiments in vivo are necessary in the future to explore the toxicological outcome of competitive interaction between AFB1 and quercetin with HSA.

Round 2

Reviewer 1 Report

Authors thoroughly answered to all comments and made the required changes in the text. Just several minor comments:

Line 56: please, replace “flavonoids” with “flavonoid” (no plural).

Line 59: please, remove the word “and” (“…ultrafiltration studies, etc.”)

Line 60: “of quercetin to competitively bind…” will be better.

Line 137: it seems that there should be a comma instead of the dot before “It could not…”

Lines 182-183, 207-208, 215-216: there is no need to repeat the same phrase prior each table. I suggest you can add the phrase “Here and in tables below, the 1st and 4th columns represent different system conditions. The other columns represent the value of fluorescence intensity of HAS” to the legend of the Table 3 with the corresponding removal of the repeated phrases from the text of the manuscript (lines 182-183, 207-208, 215-216).

Line 325,326: please, replace “from” with “within”: varied within 0-8 uM and so on. If you use the word “from”, then it should be “from 0 to 8”…

Author Response

According to the comments of reviewer, a corresponding revision of my manuscript was done. All of the changes were marked in the revised manuscript in red. The details about revised manuscript were the followings.

Reviewer 2 Report

Most of the comments from previous revision round were tackled by the authors and significant changes were made on the manuscript to improve its quality and cleanliness.

I still consider that English should be reviewed again and improved through all the document. Some examples are:

Line 15-16: this sentence is difficult to understand, please modify it improving English.

Line 48-49: this was not experimentally proved yet. It is an hypothesis, therefore the sentence should be written in conditional. Otherwise, give a reference for this statement.

Line 136-137: there is something missing

Other comments are:

- Please indicate in the figure captions the meaning of the axes titles

- Tables 3, 4 and 5 is difficult to understand, please modify it to make it easier for readers

Author Response

(The authors gave the same response as above.)
